# Genome-wide association study identifies two risk loci for tuberculosis in Han Chinese

Ruijuan Zheng[1], Zhiqiang Li[2,3], Fusheng He[4], Haipeng Liu ⬤ [1], Jianhua Chen[3], Jiayu Chen[5], Xuefeng Xie[4], Juan Zhou[3], Hao Chen[6], Xiangyang Wu[6], Juehui Wu[6], Boyu Chen[3], Yahui Liu[3], Haiyan Cui[7], Lin Fan[7], Wei Sha[7], Yin Liu[8], Jiqiang Wang[3], Xiaochen Huang[1], Linfeng Zhang[5], Feifan Xu[9], Jie Wang[1], Yonghong Feng[1], Lianhua Qin[1], Hua Yang[1], Zhonghua Liu[1], Zhenglin Cui[1], Feng Liu[1], Xinchun Chen[10], Shaorong Gao ⬤ [5], Silong Sun[4], Yongyong Shi ⬤ [2,3,11] & Baoxue Ge[1,6]

Tuberculosis (TB) is an infectious disease caused by *Mycobacterium tuberculosis* (*Mtb*), and remains a leading public health problem. Previous studies have identified host genetic factors that contribute to *Mtb* infection outcomes. However, much of the heritability in TB remains unaccounted for and additional susceptibility loci most likely exist. We perform a multistage genome-wide association study on 2949 pulmonary TB patients and 5090 healthy controls (833 cases and 1220 controls were genome-wide genotyped) from Han Chinese population. We discover two risk loci: 14q24.3 (rs12437118, $P_{combined} = 1.72 \times 10^{-11}$, OR = 1.277, *ESRRB*) and 20p13 (rs6114027, $P_{combined} = 2.37 \times 10^{-11}$, OR = 1.339, *TGM6*). Moreover, we determine that the rs6114027 risk allele is related to decreased *TGM6* transcripts in PBMCs from pulmonary TB patients and severer pulmonary TB disease. Furthermore, we find that *tgm6*-deficient mice are more susceptible to *Mtb* infection. Our results provide new insights into the genetic etiology of TB.

[1] Shanghai Key Lab of Tuberculosis, Shanghai Pulmonary Hospital, Tongji University School of Medicine, Shanghai 200043, China. [2] The Affiliated Hospital of Qingdao University &The Biomedical Sciences Institute of Qingdao University (Qingdao Branch of SJTU Bio-X Institutes), Qingdao University, Qingdao 266003, China. [3] Bio-X Institutes, Key Laboratory for the Genetics of Developmental and Neuropsychiatric Disorders (Ministry of Education) and the Collaborative Innovation Center for Brain Science, Shanghai Jiao Tong University, Shanghai 200030, China. [4] BGI Genomics, BGI-Shenzhen, Shenzhen 518083, China. [5] Clinical and Translational Research Center of Shanghai First Maternity and Infant Hospital, Shanghai Key Laboratory of Signaling and Disease Research, School of Life Science and Technology, Tongji University, Shanghai 200092, China. [6] Department of Microbiology and Immunology, Tongji University School of Medicine, Shanghai 200092, China. [7] Tuberculosis Center for Diagnosis and Treatment, Shanghai Pulmonary Hospital, Tongji University School of Medicine, Shanghai 200043, China. [8] Department of Clinical Laboratory, Shanghai Pulmonary Hospital, Tongji University School of Medicine, Shanghai 200043, China. [9] Clinical Laboratory, The Sixth People's Hospital of Nantong, Jiangsu 226011, China. [10] Shenzhen University School of Medicine, Shenzhen 518060, China. [11] Department of Psychiatry, First Teaching Hospital of Xinjiang Medical University, Urumqi 830054, China. These authors contributed equally: Ruijuan Zheng, Zhiqiang Li, Fusheng He. Correspondence and requests for materials should be addressed to H.L. (email: haipengliu@tongji.edu.cn) or to S.S. (email: sunsilong@genomics.cn) or to Y.S. (email: shiyongyong@gmail.com) or to B.G. (email: gebaoxue@sibs.ac.cn)

Tuberculosis (TB), an infectious disease caused by *Mycobacterium tuberculosis* (*Mtb*), remains the leading cause of death by infection in the world[1], with 10.4 million new cases and 1.7 million deaths in 2016[2]. Substantial evidence from twin[3,4], family linkage[5], and candidate gene studies[6] has demonstrated that host genetic factors are key players in determining the susceptibility to TB.

Four common variants associated with TB have been identified through genome-wide association studies (GWAS) in African and/or Russian populations[7-9]. However, some of these variants have much lower frequencies in other populations, making the data difficult to replicate[10-13]. Besides, the human leukocyte antigens (HLA) class II region was found to contribute to genetic risk of TB in Europeans[14]. Another GWAS performed in an Asian population did not identify susceptibility loci that reached the threshold for genome-wide significance[15,16], likely because of smaller sample sizes. As much of the heritability in TB remains unaccounted for and more susceptibility loci for TB likely remain to be identified, it is valuable to perform GWAS for TB among different ethnic groups to further understanding the genetic basis of TB.

To identify the loci for susceptibility to TB, we perform a three-stage GWAS in the Han Chinese population using 2949 pulmonary TB cases and 5090 controls. We identify two susceptibility loci that are significantly associated with TB at 14q24.3 (rs12437118, $P = 1.72 \times 10^{-11}$ for *ESRRB*) and 20p13 (rs6114027, $P = 2.37 \times 10^{-11}$ for *TGM6*). Moreover, we find the rs6114027 regulates the expression of the *TGM6* gene and is associated with the severity of TB. Furthermore, by generating *tgm6*-deficient mice we demonstrate that *tgm6* protects mice from *Mtb* infection. Taking together, our findings provide shed light on the genetic etiology and pathogenesis of TB.

## Results

**GWAS and replications**. To identify susceptibility loci for TB, we performed a three-stage GWAS of TB in a Han Chinese population. We conducted genome-wide genotyping of 2112 subjects using Affymetrix Axiom CHB arrays (discovery stage). After quality control (QC) and imputation, a total of 5,374,021 variants with a minor allele frequency (MAF) of 3% or greater in 833 pulmonary tuberculosis cases and 1220 controls were analyzed (Supplementary Table 1). We utilized principal component analysis (PCA) to detect potential population stratification in our samples (Supplementary Figure 1) and conducted logistical regression analysis adjusted for the first three principal components in the discovery stage. After PCA-adjustment, we did not observe significant inflation of test statistics (inflation factor $\lambda = 1.01$, Supplementary Figures 2 and 3), suggesting that effects from potential population stratification were well-controlled.

In the discovery stage, none of the single nucleotide polymorphisms (SNPs) achieved the genome-wide significance threshold ($P$ value $= 5 \times 10^{-8}$) for the Affymetrix Axiom®

Genome-Wide CHB1 and CHB2 Array Plate Set (two chips/set). We genotyped the lead SNPs or their proxies from the top ten most significantly associated independent loci ($P < 1 \times 10^{-5}$, Supplementary Data 1) in two independent sample sets. Replication sample set 1 was comprised of 1074 cases and 1904 controls; replication sample set 2 was comprised of 1042 cases and 1966 controls (Supplementary Table 1). The combined analysis of the discovery and replication stages gave a total sample size of 2949 cases and 5090 controls, which can provide sufficient statistical power (>85%) to detect associations with genotypic relative risks (GRR) of 1.30 for variants with risk-allele frequencies (RAF) of ~20 to 75% (Supplementary Table 2). In the replication stage, six SNPs had signals with the same direction of effect as was observed in the discovery analysis, and three of them showed significance ($P < 0.05$) (Supplementary Table 3). In the combined analysis two SNPs (rs12437118 at 14q24.3 and rs6114027 at 20p13) were genome-wide significantly associated with TB. Both SNPs showed significance ($P < 0.05$) in all the three data sets and were consistent across data sets (Table 1). We also queried epigenetic annotations for 14q24.3 and 20p13 (indexed by rs12437118 and rs6114027, respectively) to explore potential implications of the association signals. The index SNPs and most of their linked variants ($r^2 > 0.2$) are predicted to have putative regulatory function (Supplementary Data 2).

The most significant signal was observed at rs12437118 ($P_{\text{combined}} = 1.72 \times 10^{-11}$, odds ratio (OR) = 1.277, Fig. 1a) at 14q24.3, the allele frequencies for risk allele A ranged from 0.261 to 0.288 in the three stages, and no heterogeneity across the stages was found ($P_{\text{het}} = 0.719$ and heterogeneity index $I^2 = 0$). SNP rs12437118 is located in an intergenic region 16 kb downstream of *ESRRB*. Common variations within this region have been associated with autosomal-recessive nonsyndromic hearing impairment[17], acute lymphoblastic leukemia (childhood)[18], and healing of the rotary cuff[19].

The second significant signal was observed at rs6114027 ($P_{\text{combined}} = 2.37 \times 10^{-11}$, OR = 1.339, Fig. 1b) at 20p13, the allele frequencies for risk allele C ranged from 0.153 to 0.160 in the stages, and no heterogeneity was observed ($P_{\text{het}} = 0.399$ and $I^2 = 0$). SNP rs6114027 is located in the intron region of the *TGM6* gene. The protein encoded by *TGM6* belongs to the transglutaminase superfamily that catalyzes cross-linking of proteins and conjugation of polyamines to proteins. Mutations in this gene have been reported to be associated with spinocerebellar ataxia type 35 (SCA35)[20] and acute myeloid leukemia[21].

We investigated the association of the SNPs identified in the published TB GWAS[7-9,14] with susceptibility to TB based on our data set. Four SNPs located outside the HLA region were highlighted in previous reports[7-9]: rs4733781 and rs10956514 in the *ASAP1* gene and rs2057178 on chromosome 11p13 as well as rs4331426 on chromosome 18q11.2. For all these SNPs the direction of the effects analyzed in our GWAS were consistent with published findings. For rs2057178 and rs4331426, the OR values are comparable for our discovery set as well as previous

**Table 1 Genome-wide significant association of two independent SNPs with TB**

| SNP | Phase | Effect allele (freq.) | OR [95% CI] | P value |
|---|---|---|---|---|
| rs12437118 | Discovery | A (0.288) | 1.326 [1.154-1.524] | 7.06E−05 |
| Chr14:76983730 | Replication 1 | A (0.269) | 1.292 [1.149-1.452] | 1.72E−05 |
| *ESRRB* | Replication 2 | A (0.261) | 1.228 [1.092-1.381] | 5.91E−04 |
| | Combined | | 1.277 [1.189-1.371] | 1.72E−11 |
| rs6114027 | Discovery | C (0.153) | 1.480 [1.244-1.760] | 9.37E−06 |
| Chr20:2379598 | Replication 1 | C (0.160) | 1.285 [1.116-1.479] | 4.92E−04 |
| *TGM6* | Replication 2 | C (0.158) | 1.306 [1.138-1.499] | 1.45E−04 |
| | Combined | | 1.339 [1.229-1.459] | 2.37E−11 |

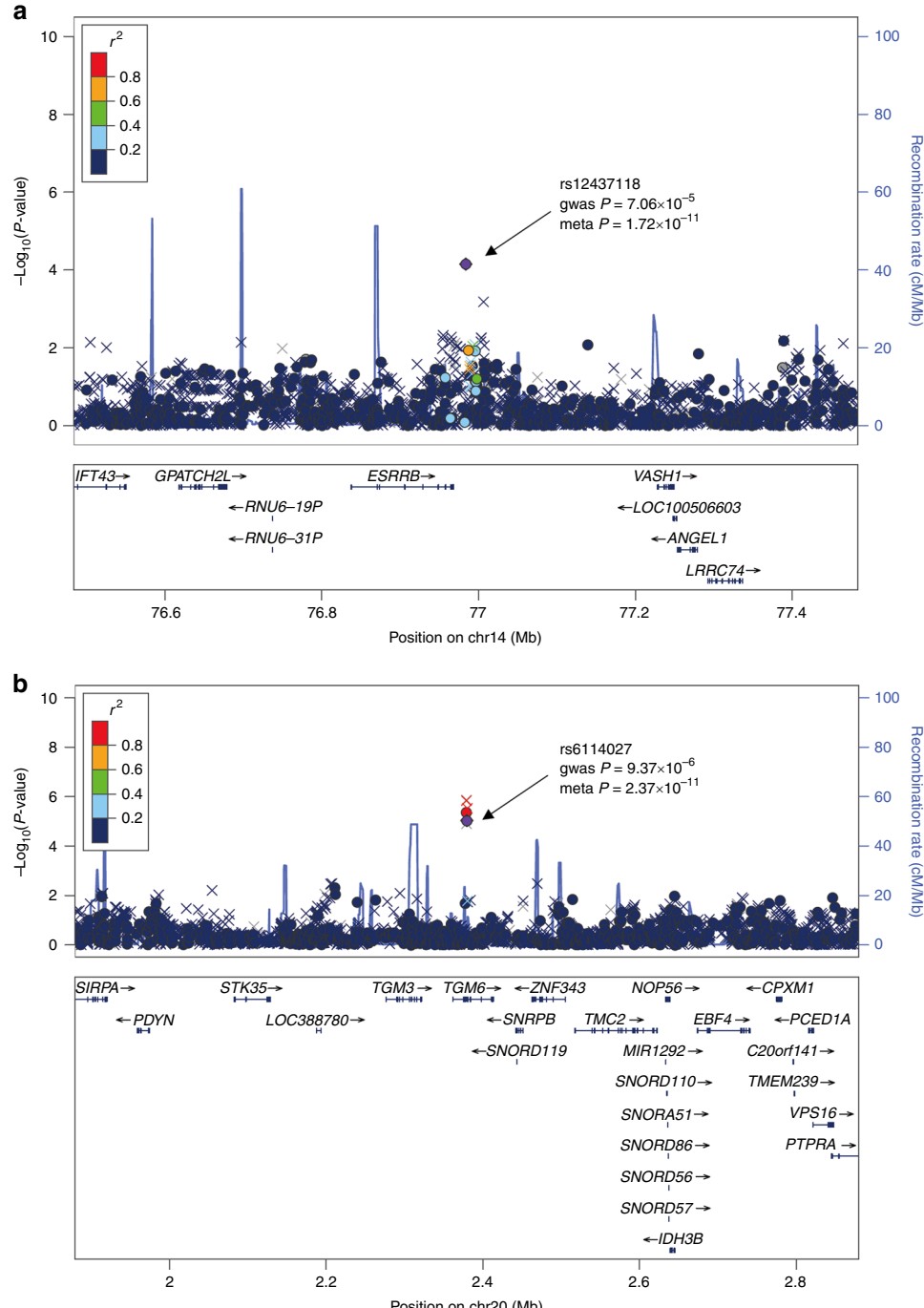

**Fig. 1** Regional association plots of loci associated with TB. **a** rs12437118 and **b** rs6114027. Purple circles represent the marker SNP in each region in the meta-analysis of discovery and replication. −log$_{10}$ P values (y axis) of the SNPs (within the regions spanning 500 kb on either side of the marker SNP) are presented according to the chromosomal positions of the SNPs (x axis, hg19). Genotyped and imputed SNPs are shown with circle and cross symbols, respectively. SNPs are shown in different colors based on their linkage disequilibrium (LD) with the marker SNP, which were established based on the 1000 Genome Asian (ASI) data (March 2012). Recombination rates estimated from the 1000 Genomes Project March 2012 samples are shown as blue lines, and within the interested regions, the genomic locations of genes annotated from the UCSC Genome Browser are displayed as arrows

studies (0.88 vs. 0.82 and 1.19 vs. 1.19, respectively). However, none of the SNPs achieved the genome-wide significance threshold in our data set (Supplementary Table 4), which might be due to the rarity (<5%) of these SNPs in Chinese populations as well as the limited sample size of our discovery data set. Three SNPs (rs557011, rs9271378, and rs9272785) in the HLA region were found to be genome-wide significantly associated with TB phenotypes in Europeans[14]. We observed nominal significant

association with TB risk in Chinese samples for rs557011 and rs9271378 (located between *HLA-DQA1* and *HLA-DRB1*, P < 0.05 and with a consistent direction of effect, Supplementary Table 4), but failed to replicate the association for rs9272785 (located in *HLA-DQA1*). We also imputed HLA alleles of *HLA-A*, *HLA-B*, *HLA-C*, *HLA-DPA1*, *HLA-DPB1*, *HLA-DQA1*, *HLA-DQB1*, and *HLA-DRB1* using SNP2HLA[22] with a Han Chinese reference panel[23]. Like rs9272785, our data did not support the

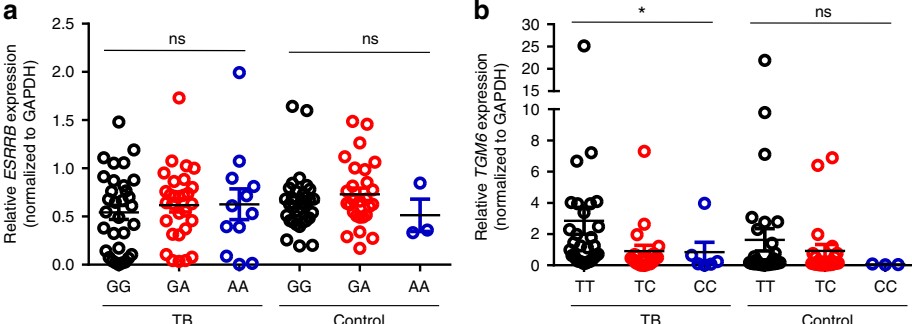

**Fig. 2** Association of SNPs with mRNA expression in PBMCs. Genomic DNA and mRNA were isolated from PBMCs from 73 TB patients (TB) and 61 healthy individuals (Control). *ESRRB* and *TGM6* mRNA expression was measured via quantitative RT-PCR and normalized to *GAPDH*. **a** The association between *ESRRB* mRNA expression and rs12437118 genotype. **b** The association between *TGM6* mRNA expression and rs6114027 genotype. Each symbol represents an individual. Data are representative of one independent experiments. Data shown were mean ± SD and statistical significance was determined via linear regression, *$P < 0.05$; ns, not significant

HLA-DQA1*03:01 (the most significantly associated HLA allele in Europeans)[14] as a risk factor for TB in Chinese (Supplementary Data 3). None genome-wide significant association was observed for all the tested HLA alleles (Supplementary Data 3).

We further conducted complementary gene and pathway-based association analyses for leveraging our genome-wide data set. None of the tested genes or pathways reached significance after Bonferroni correction ($P < 2 \times 10^{-6}$). In the gene-based analysis, the most associated gene was *BMP3* ($P = 9.53 \times 10^{-5}$, Supplementary Data 4). In addition, two other biologically plausible genes for TB (*IL18R1* and *IL1RL1*) were also found in the top ranked gene list ($P = 5.24 \times 10^{-4}$ and $6.77 \times 10^{-4}$, respectively). Of the top five most significant gene sets in our pathway-based analysis, four are adhesion related ($P = 4.83 \times 10^{-5}$ to $6.79 \times 10^{-6}$, Supplementary Data 5).

**Functional assessments.** To explore the potential biological function of rs12437118 and rs6114027, the expression quantitative trait locus (eQTL) mapping was conducted in peripheral blood mononuclear cells (PBMCs) of 73 TB patients and 61 healthy individuals. The results showed that the rs12437118 was not associated with *ESRRB* mRNA expression in PBMCs (Fig. 2a). For rs6114027, C allele carriers with a higher risk for TB had significantly lower *TGM6* expression in PBMCs than non-C allele carriers harboring a lower TB risk (Fig. 2b). SNP rs6114027 is located in the intron between exon 5 and 6 of the *TGM6* gene (Supplementary Figure 4a). We then interrogated whether the intron region including the allele affects *TGM6* gene expression by function as an enhancer or suppressor. To this end, we generated a promoter reporter construct containing a 1514 bp intron fragment with the SNP rs6114027 T or C allele (Supplementary Figure 4b) and performed a dual-luciferase assay. We found that the C allele inhibited luciferase expression in HEK293T cells to a significantly greater extent as compared with the T allele, suggesting this SNP may affect *TGM6* expression by modulating the suppressive effect of the intron on gene transcription (Supplementary Figure 4c).

We further analyzed the association of polymorphism of rs6114027 with clinical indices including the plasma levels of cytokines (IL-6, sIL-2R, and TNF-α), erythrocyte sedimentation rate (ESR) and the presence of cavity in the lung as well as the presence of mycobacteria in the sputum smear (Supplementary Figure 5a). The abundance of plasma IL-6 was significantly lower in the patients with rs6114027CC genotype than those with rs6114027TT genotype (Supplementary Figure 5b). Moreover, TB patients carrying rs6114027CC genotype showed significantly higher level of plasma sIL-2R than those with rs6114027TT

genotype (Supplementary Figure 5c). However, the abundance of plasma TNF-α and ESR were comparable among TB patients carrying different rs6114027 genotypes (*TT*, *TC*, and *CC*) (Supplementary Figure 5d, e). Of note, TB patients carrying rs6114027CC genotype showed significantly higher frequencies of mycobacteria in the sputum smear, but no difference in the lung cavity, than those with rs6114027TT genotype (Supplementary Figure 5f, g), indicating that the rs6114027C allele is a predisposing factor for severer pulmonary TB disease.

To assess the role of *TGM6* in the pathogenesis of TB in vivo, we generated *tgm6*-deficient mice by CRISPR/Cas9-mediated genome editing[24,25] (Supplementary Figure 6). We found that *tgm6* homozygous knockout mice are not lethal (Supplementary Figure 7a). There are no apparent differences in the gross phenotype such as body weight among the wild type, *tgm6* heterozygous and homozygous mice (Supplementary Figure 7a, b). All three genotypes of mice were employed for *Mtb* infection intraperitoneally. Both histopathological impairments and bacterial burden in the lung were examined. The result demonstrated that *tgm6*-deficient mice had the most robust infiltration of immune cells and inflammatory lesions among all three genotypes, indicating that *Mtb* infection led to severest tissue damage in the lung of *tgm6*-deficient mice (Fig. 3a, b). The bacterial burdens in the lung tissue as detected by both acid-fast staining and CFU assay were higher in *tgm6*-deficient mice than those in either heterozygous or wild type mice (Fig. 3c, d). Furthermore, we measured the expression of cytokines and chemokines in the lung of infected mice and found *tgm6*-deficient mice harbored the highest levels of *il-10*, *tnf-α*, and *ccl-3* mRNA among all three genotypes (Fig. 3e). Taken together, *tgm6* confers protection of mice against *Mtb* infection likely in dose-dependent manner.

## Discussion

In this three-stage GWAS of TB, we identified two loci on 14q24.3 derived from *ESRRB* and on 20p13 derived from *TGM6* that were significantly associated with TB risk in a Chinese Han population. We further demonstrated that the rs6114027C allele was associated with lower expression levels of *TGM6* in the PBMCs of TB patients, and *tgm6* deficiency rendered increased susceptibility to *Mtb* infection in mice.

The top signal SNP (rs12437118) is localized in an intergenic region 16 kb downstream of the estrogen-related receptor beta (*ESRRB*) gene. ESRRB is a member of the orphan nuclear receptor subgroup of the nuclear receptor superfamily[26]. It has been reported that the *ESRRB* gene is expressed in all human tissues, including the lung[17]. ESRRB plays a critical role in the

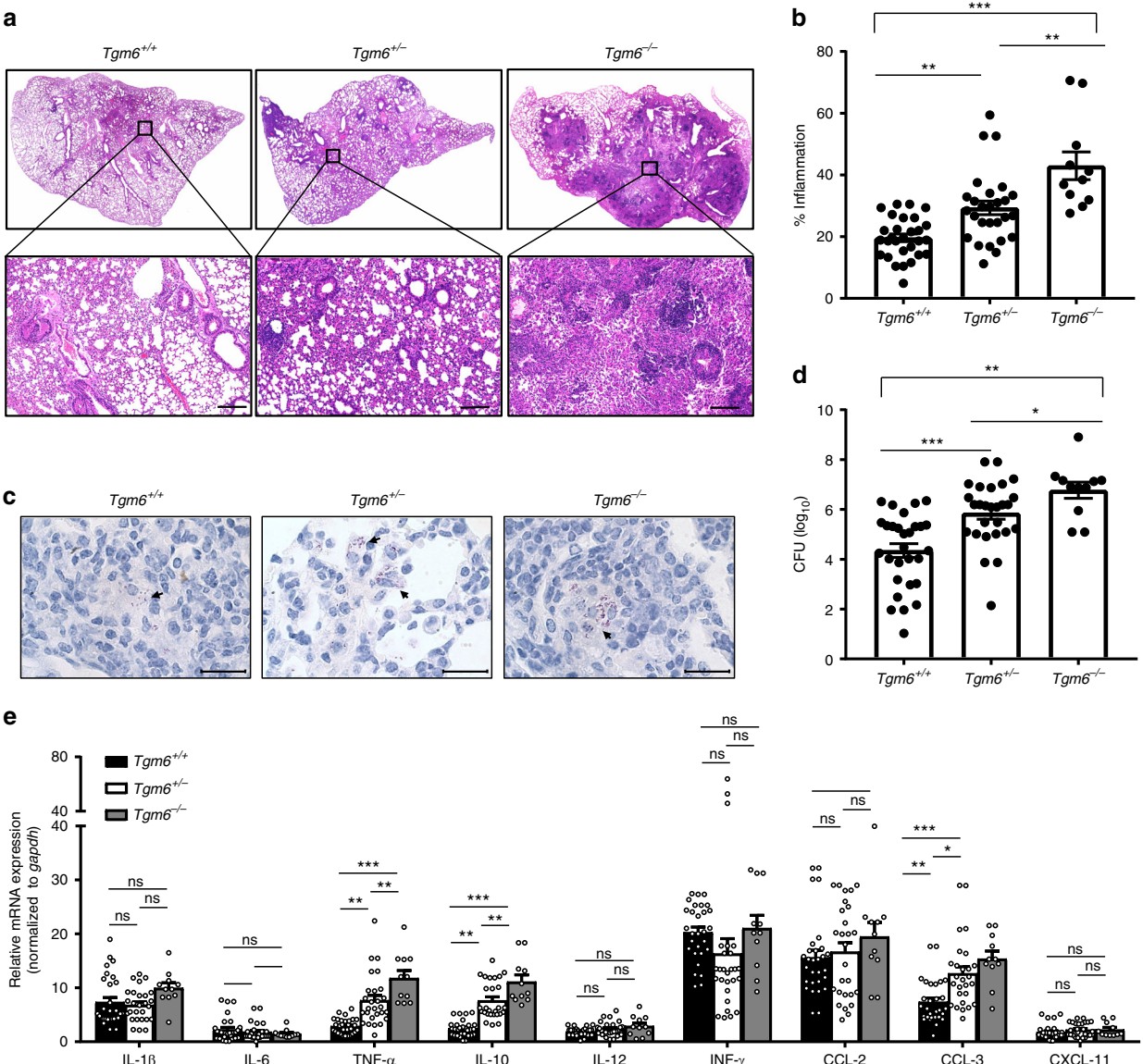

**Fig. 3** $Tgm6^{-/-}$ mice are more susceptible to $Mtb$ infection. **a** H&E staining of the lungs from $Mtb$-infected mice of indicated genotype. The upper row showing the representative picture of the whole lobe of the stained lung from mice. The lower row showing the enlarged vision of the selected region. Scale bar, 200 μm; magnification: ×100. **b** Results showing the quantification of the percentages of inflammation regions as measured by H&E staining in one lobe of the lung from mice of indicated genotype. Data shown are mean ± SEM. One-way ANOVA followed by the Bonferroni post hoc test were used for statistical analyses. **c** Representative results of acid-fast staining of $Mtb$ in the lungs from mice infected with $Mtb$. Scale bar, 20 μm; magnification: ×1000. **d** The bacterial burden in the lungs from $Mtb$-infected mice was measured by a CFU assay. Data were presented as the medians ± interquartile ranges and were analyzed by the Mann–Whitney $U$ test. **e** Quantitative RT-PCR measurement of the mRNA levels (normalized to $gapdh$) of cytokines and chemokines in lungs from infected mice of indicated genotypes. The data for $tgm6^{+/+}$ ($n = 29$) and $tgm6^{+/-}$ ($n = 27$) mice were pooled from four independent experiments and the data for $tgm6^{-/-}$ mice ($n = 11$) were pooled from two independent experiments. Data shown were mean ± SEM. One-way ANOVA followed by the Bonferroni post hoc test were used for statistical analyses. *$P < 0.05$; **$P < 0.01$; ***$P < 0.001$; ns, not significant

development of several tissues and organ systems and $esrrb$ knockout mice die at 10.5 days postcoitum[27]. ESRRB is involved in the development of the inner ear and hearing;[28] conditional $esrrb^{-/-}$ mice show a defect in vestibular function and hearing impairment. In humans, mutations in the $ESRRB$ gene were found in autosomal-recessive nonsyndromic hearing impairment DFNB35[29,30]. Until now, the function of $ESRRB$ in TB infection was unknown. A previous study indicated that ESRRB directly bound to $Gata6$ promoter and resulted in maintenance of the undifferentiated status of embryonic stem cells[31]. Interestingly, $GATA6$ was previously thought to be a TB susceptible gene due to close proximity to a susceptibility locus for TB on chromosome

18q11.2 variant (SNP rs4331426) identified by GWAS[7]. $GATA6$ is required for both lung development and regeneration by balancing stem/progenitor expansion and epithelial differentiation via Wnt signaling[32]. Our GWAS analyses determined that the $ESRRB$ gene was significantly associated with TB. This make it tempting to hypothesize that ESRRB contributes to TB pathogenesis by regulating the regeneration of damaged lung tissue caused by $Mtb$ infection, which warrants additional studies. However, no difference was detected between rs12437118 and $ESRRB$ mRNA in PBMCs. It is, therefore, highly likely that the differences in ESRRB abundances across the different genotypes may be restricted to some other tissues.

The second identified SNP, rs6114027, was mapped to the intron region of the *TGM6* gene. The evidence suggests that *TGM6* plays an important role in neurologic diseases; however, the exact mechanism of action remains elusive. Via exome sequencing and family linkage analysis, *TGM6* has been identified as a causative gene of spinocerebellar ataxias[20], and acute myeloid leukemia[21]. Several identified mutations of *TGM6* gene were reported to be prone to cause death of primary neurons resulting from unfolded protein reaction activation, which may contribute to pathogenesis of SCA35[33]. As an autoimmune target, anti-TGM6 antibodies were presented at a higher level in patients with progressive multiple sclerosis (MS)[34], amyotrophic lateral sclerosis[35], and gluten-sensitive cerebellar ataxia[36,37]. In this study, we identify a mutation in *TGM6* associated with TB; our eQTL analysis showed that the rs6114027C allele was associated with significantly lower expression levels of *TGM6* in PBMCs from TB patients. The luciferase assay further validated a suppressive effect of SNP rs6114027C allele on gene transcription. Most risk alleles have been found to alter transcription–regulatory element binding[38], however, whether this intronic SNP regulates gene expression via modulating suppressor activity warrants further investigation.

Here, we report an important role of TGM6 in the process of *Mtb* infection. Aerosol route using inhalation exposure system is more physiologic and relevant to natural infection, which is more favored for *Mtb* infection in mouse model. However, due to the inaccessibility to the aerosol infection model, we employed an intraperitoneal injection model, which has been reported previously[39–41]. We successfully generated a strain of tgm6 knockout mice with a protein null mutant by CRISPR/Cas9-mediated genome editing. No gross phenotypic defects were observed in both tgm6 heterozygous and homozygous mice. However, tgm6 deficiency led to a higher bacterial burden and more pathological impairments in the lung of mice post *Mtb* infection. Moreover, we observed that the deletion of tgm6 resulted in elevated transcripts of il-10, tnf-α, and ccl-3 mRNA in the lung of *Mtb*-infected mice. However, whether the enhanced cytokines response is secondary to the increased bacterial burden or directly modulated by *TGM6* warrants further investigation.

By analyzing the association of polymorphism of rs6114027 with clinical indices, we found that the rs6114027C allele is associated with severer pulmonary TB disease as manifested with higher frequencies of mycobacteria in the sputum smear. Notably, the elevated *TGM6* in astrocytes was found to regulate the expression of glial fibrillary acidic protein (GFAP) in progressive MS[34]. As a marker of astrocyte activation, GFAP reflects the level of damage to the blood–brain barrier to some extent. Of note, GFAP is highly expressed in the brain of tuberculosis meningitis, a fatal form of *Mtb* infection of the central nervous system[42]. Therefore, how *TGM6* regulates the dissemination of *Mtb* and development of severe TB such as extrapulmonary TB remains an interesting question to be addressed.

In the gene-based analysis, the top gene associated with TB was bone morphogenetic protein 3 (*BMP3*). *BMP3* is a member of the transforming growth factor-beta superfamily and regulates osteoblast differentiation. To identify TB-risk associated biological pathways, we performed pathway-based association analysis. Of note, among these identified pathways, the top four pathways were related to adhesion including the adherens junction organization pathway, the focal adhesion assembly pathway, the cell substrate adherens junction assembly pathway as well as the adherens junction assembly pathway. These pathways were important for the regulation of hematopoietic progenitor cells and were essential for both immune and inflammatory responses. It has been reported mycobacterium infection decreased β-catenin expression, which is an adherens junction protein that promotes cell-cell adhesion and contributes to the control of pleural permeability, leading to TB pleurisy[43].

The newly identified TB associated SNPs in Chinese (rs12437118 and rs6114027) also are common (MAF: 0.10–0.35, Supplementary Table 5) in African and European populations, but our findings were not seen previously[7–9,14]. Such discrepancy in GWAS findings on the susceptibility loci to TB across populations have been encountered previously. The *ASAP1* locus identified in Russian[9] is failed to be replicated in Icelandic[14], and Chinese individuals. The inconsistence in GWAS may be due to different causal variants or linkage disequilibrium structures in different populations, or phenotypic heterogeneity. Of note, as a complex infectious disease, TB is an outcome of the intricate interaction between genome and environment. The putative environmental influences may result in genome-wide epigenetic modification, which in turn regulates the pathways relevant to TB pathogenesis in cooperation with genetic variation and gene expression[44]. The validation of a protective effect of TGM6 in the process of *Mtb* infection in mice indicates the reliability of our GWAS study in Han Chinese population.

Taken together, our GWAS in a Chinese population has identified two genetic variants of the *ESRRB* and *TGM6* genes that are associated with susceptibility to TB in Chinese Han population. Our results provide the insight into the genetic etiology of TB and further studies are required to explore how *ESRRB* and *TGM6* are involved in the regulation of *Mtb* infection and in the predisposition to TB.

## Methods

**Ethics**. All of the participants provided written informed consent. The study was conducted according to the Declaration of Helsinki principles. The protocol was approved by the local ethics committee of Tongji University School of Medicine and signed informed consent was obtained from all subjects (permit number: 2011-FK-03, No. 2015YXY48).

**Samples**. We performed a three-stage case–control analysis, including an initial discovery stage and two stages of follow-up. TB patients and controls were collected at three Chinese cities: Shanghai (municipality), Nantong (Jiangsu province), and Shenzheng (Guangdong province), according to the same protocol. In the discovery stage, 833 pulmonary tuberculosis patients and 1220 healthy controls recruited from Shanghai Pulmonary Hospital affiliated with Tongji University, were genotyped using an Axiom Genome-Wide CHB1 & CHB2 Array Plate Set (two chips/set, Affymetrix). During the replication stage, the first-stage validation samples were collected from patients at Shanghai Pulmonary Hospital affiliated with Tongji University (1074 cases and 1904 controls); second-stage validation samples were collected from patients at Shanghai Pulmonary Hospital affiliated with Tongji University (510 cases and 1190 controls), the Sixth People's Hospital of Nantong (297 cases and 398 controls), and Shenzhen Third People's Hospital (235 cases and 378 controls).

Diagnostic criteria for TB were as follows: (1) sputum culture for *Mtb*; (2) presence of acid-fast bacilli in sputum smear; (3) clinical presentation and radiological signs (such as X-ray or computed tomography scan). The diagnosis all TB patients was ultimately confirmed by culture of *Mtb* from sputum. Sputum samples were cultured on Löwenstein–Jensen (L.J.) culture media and a BACTEC MGIT 960 system. Identification of the *M. tuberculosis* complex (MTBC) was performed by the conventional p-nitrobenzoic acid (PNB) and thiophene carboxylic acid hydrazine (TCH) resistance tests. Growth in LJ medium containing PNB indicated that the bacilli did not belong to the MTBC. The patients with extrapulmonary TB were excluded. The flow chart of the sample collection of TB patients was listed in Supplementary Figure 8. The total 3044 culture-positive samples were collected for genomic DNA extraction.

For analysis of association between genetic variation of *TGM6* SNP rs6114027 and clinical indices, the clinical data of the samples involved in the discovery stage and the first replication stage were collected, including the plasma levels of cytokines (IL-6, sIL-2R, and TNF-α), ESR, and the presence of cavity in the lung as well as the presence of mycobacteria in the sputum smear. The sample treated with anti-TB treatment were excluded. The characteristics of the patients were shown in Supplementary Table 6 and the design of the study was in Supplementary Figure 5a.

The controls were recruited from a pool of individuals who participated in a health examination program and were living mainly in the same geographical area as the cases. The control populations were also subject to the physical examination, blood testing, and chest X-rays (CXR) and were screened for a history of TB and a family history of TB; those subjects with signs, symptoms, and CXR results

suggestive of active TB, a history of previous TB, or anti-TB treatment were excluded. *Mtb* infection status of these controls was unknown. All cases and controls with alcoholism, diabetes, chronic use of corticosteroids, or immunodeficiency were excluded. Controls were not tested for antibodies to human immunodeficiency virus (HIV), but TB cases were tested and excluded if HIV-positive.

**Genomic DNA extraction.** Venous blood samples were collected with ethylene-diaminetetraacetic acid disodium salt (EDTA-2Na)-anticoagulant tubes from all participants, followed by genomic DNA extraction using Flexi Gene DNA kits (Qiagen) and the QuickGene DNA whole-blood kit (Fujifilm) according to the manufacturer's instructions[45]. The extracted DNA was diluted to working concentrations of 50 ng/µl for genome-wide genotyping, and 15–20 ng/µl for the validation study.

**Genome-wide genotyping and QC.** An Affymetrix Axiom® Genome-Wide CHB1 & CHB2 Array Plate Set (two chips/set) was used for genome-wide genotyping in the discovery stage. The QC analysis and genotype calling were performed for the CHB1 and CHB2 arrays separately according to the Axiom® Genotyping Solution Data Analysis Guide. Chips with a dish QC (DQC) value less than 0.82 were excluded in the primary QC step. Samples with passing DQC values underwent step 1 genotype calling using a subset of probe sets. Samples with a call rate <97% or in a nonpassing plate (the average call rate of passing samples <98.5%) were then excluded. The post-QC samples were kept for Step 2 genotype calling using the Axiom Genotyping Algorithm v1 (Axiom GT1). The SNP polisher was utilized for SNP QC and SNPs in the recommended categories (PolyHighRes, MonoHighRes, and No Minor Homand Hemizygous) were kept, with visual inspection if required or advised.

Next, estimated gender via genotyping data was compared with the sample record, sample with inconsistent gender were removed. Heterozygosity rates were calculated with the intent of removing deviations that exceeded six standard deviations from the mean. PLINK's identity-by-descent analysis was used to detect cryptic relatedness. When a pair of individuals exhibited a PI_HAT > 0.2, the member of the pair with the lower call rate was excluded from the analysis. SNPs with call rates <97%, MAF < 3%, or significant deviations from the Hardy–Weinberg equilibrium (HWE) in the controls (HWE, $P \leq 1 \times 10^{-6}$) were excluded. Ungenotyped SNPs in the GWAS discovery samples were imputed using SHAPEIT 2.0[46] (phasing step), IMPUTE2[47] (imputation step) and haplotype information from the 1000 Genomes Project (Phase I integrated variant set across all 1092 individuals, v2, March 2012; see URLs). The variants with INFO > 0.8 were saved for further analysis, and we also applied the same SNP QC criteria as the aforementioned genotyped analysis. A total of 5,374,021 SNPs passed the quality criteria and were used in the subsequent analyses.

**SNP selection and genotyping in replications.** We selected independent SNPs with PCA-adjusted and/or unadjusted *P* values less than $1.0 \times 10^{-5}$ for the replication study. Genotyping for replication was performed using the iPLEX platform (Sequenom, San Diego, CA, Supplementary Data 6). To exclude possible genotyping error, the genotype cluster plots of the selected SNPs for the discovery and replication stages were generated and visually inspected (Supplementary Figures 9 and 10).

**Statistical analysis.** Population stratification analysis was conducted using a PCA-based method implemented in the software package EIGENSTRAT[48]. Power estimates were calculated for the total sample size used in our study with the GAS Power Calculator[49], giving a range of GRR and RAF and assuming a population prevalence of 0.004[50], and a significance level of $5 \times 10^{-8}$[51]. Logistic regression was used to test the association of a single SNP using PLINK[52], and the top three principal components were used as covariates in the association analysis to correct for the population stratification. The independent effects of an individual SNP were evaluated using conditional analysis. In the meta-analysis, we adopted inverse variance weighting fixed-effects model to combine the results from different datasets, and used Cochran's *Q* test and the $I^2$ index to quantify the degree of heterogeneity across the datasets. A Manhattan plot of the $-\log_{10}$ (*P* values) was generated using Haploview[53]. Regional plots were generated using the online tool LocusZoom 1.2[54] (see URLs). The Mann–Whitney *U* test, linear regression and ANOVA followed by Bonferroni post hoc test was performed using Graph-Pad Prism 6.0 software. The genome-wide significant SNPs were annotated using HaploReg[55] based on the LD information from the Asian 1000 Genomes Project population.

**Analysis of association between SNPs and mRNA expression.** For validation of the association between genetic variation and *ESRRB* and *TGM6* mRNA expression, 73 TB patients and 61 healthy individuals were recruited. Two tubes of whole-blood samples were collected from each donor and subjected to DNA extraction and isolation of PBMCs for RNA extraction, respectively.

**Peripheral blood mononuclear cells isolation.** PBMCs were isolated from EDTA-treated whole-blood through density gradient centrifugation by Ficoll separation (Ficoll-Paque plus; Amersham Biosciences) and were then washed with PBS twice. The harvested PBMCs were applied for further experiments.

**RT-PCR analysis.** TRIzol (Invitrogen) and the ReverTra Ace® qPCR RT Kit (Toyobo, FSQ-101) were used for total RNA extraction and the reverse transcription of messenger RNA (mRNA), respectively. The relative mRNA expression of different genes were detected by quantitative real-time RT-PCR with a SYBR RT-PCR kit (Toyobo, QPK-212) and was calculated by comparison with the control gene *Gapdh* (encoding GAPDH) using the $2^{-\triangle\triangle Ct}$ method. The primers used are listed in Supplementary Table 7.

**Luciferase assay.** Human embryonic kidney epithelial cells (HEK293T; ATCC CRL-11268) were maintained in Dulbecco's Modified Eagle Medium (Hyclone) supplemented with 10% (v/v) heat-inactivated fetal bovine serum (Gibco) and 100 U/ml penicillin and streptomycin. Cell lines were tested for the absence of mycoplasma contamination by PCR. The intron between exon 5 and 6 containing 1514 bp with SNP rs6114027, allele T or C, was cloned into a luciferase reporter vector (pGL3-promoter-*TGM6*^WT and pGL3-promoter-*TGM6*^MT). HEK293T cells were plated in 48-well plates 24 h before transfection. A total of 0.1 µg luciferase reporter vector and 0.1 mg of pRL-TK control vector were co-transfected into the cells using Lipofectamine 2000 reagent (Life technology). At 24 h post-transfection, cells were harvested and lysed. The Dual-luciferase Reporter Assay System (Promega) was applied for measuring both the firefly and *Renilla* luciferase activities, and the ratio between firefly and *Renilla* luciferase activity were calculated to represent promoter activity. The experiments were performed three times with technical duplicates in each independent experiment.

**Gene- and pathway-based association analyses.** We performed gene and pathway-based association testing using MAGMA[56]. The gene *P* values were computed using an *F*-test based on a multiple linear regression model and competitive tests were utilized for gene-set analysis. We used the gene sets from the Molecular Signatures Database (MSigDB) databases, which is comprised of eight major collections, including Immune Signatures[57].

**Bacteria.** *Mtb* H37Rv (H37rv) were grown in Middlebrook 7H9 broth (Becton Dickinson, Cockeysville, MD) with 0.05% Tween-80 and 10% oleic acid-albumin-dextrose-catalase (OADC) (Becton Dickinson, Sparks, MD).

**Mice and *Mtb* infection.** *Tgm6* knockout mice was generated by the CRISPR/Cas9 method[24,25]. The *tgm6* sgRNA were designed by targeting the exon4 of *tgm6* and the sequences were listed in Supplementary Table 7. C57BL/6n female mice (7–8 weeks old) were used as embryo donors. C57BL/6n female mice were superovulated by intraperitoneally injecting with pregnant mare serum gonadotropin and human choionic gonadotophin, and then mated to C57BL/6n male mice. The fertilized embryos (zygote) were collected from oviducts and received the injection of mixed Cas9 mRNA (100 ng/µL) and sgRNA (25 ng/ul) targeting *tgm6*, followed by culture in Quinn's Advantage cleavage medium (In-Vitro Fertilization, Inc.) for about 24 h, and every 18–20 two-cell stage embryos were transferred into the oviduct of a pseudo pregnant ICR female mouse at 0.5 days postcoitus (dpc). To determine the nucleotide change of mutated alleles, DNA sequence of F0 mice was performed after TA cloning into plasmid pMD19T (TAKARA). In order to obtain F1 *tgm6* knockout mice, F0 mice were crossed with C57BL/6 and newborn generations were genotyped by the Sanger sequencing. Genomic DNA was extracted from tail tips and the PCR-based genotyping were employed. The genotyping primers sequences were listed in Supplementary Table 7. C57BL/6n mice were purchased from Shanghai Laboratory Animal Center (Shanghai, China). All mice were bred in specific pathogen-free conditions at the Laboratory Animal Center of Tongji University. Female mice (7–8 weeks old) were divided randomly into cages upon arrival and were infected by injection of $3.5 \times 10^5$ cfu *Mtb* (H37Rv) intraperitoneally (i.p.) in the Biosafety Level-3 (BSL-3) Laboratory for 28 days. The age/sex matched *tgm6* wild type littermates were used as controls. All mice experiments were performed in accordance with the University Health Guide for the Care and Use of Laboratory Animals and were approved by the Biological Research Ethics Committee of Tongji University.

**Immunoblot analysis.** The lung tissues were isolated from mouse and protein was extracted using RIPA Lysis Buffer (Beyotime, China) according to the manufacturer's instructions. The protein sample lysate was separated using 12% sodium dodecyl sulfate (SDS)-polyacrylamide gel, transferred onto Polyvinylidene Fluoride (PVDF) membrane and incubated with the appropriate antibodies. Rabbit anti-TGM6 (TA331020, 1:1000) was purchased from OriGene Technologies, Inc; rabbit anti-GAPDH (SAB2701826, 1:1000) was purchased from Sigma.

**Colony forming unit assay.** The infected mice were sacrificed at indicated time points and the lung tissues were harvested and homogenized in 5 ml PBS. Tenfold serial diluents of the lung homogenates were then plated on Middlebrook 7H10

agar supplemented with 10% Middlebrook OADC enrichment (both from Becton Dickinson, Sparks, MD) followed by incubation at 37 °C for around 3 weeks before counting the colonies.

**Histological analysis and acid-fast staining**. Following fixation with 4% phosphate-buffered formalin for 24 h, the lung tissues from *Mtb*-infected mice were embedded in paraffin wax and were cut into serial sections of 2–3 μm thickness. The infiltration of immune cells into the lungs were detected by Haematoxylin and eosin (H&E) staining. *Mtb* burden in the lungs were determined by acid-fast staining with standard Ziehl–Neelsen method. Light microscopy was applied for the visualization of the stained slides.

**URLs**. For PLINK, see http://pngu.mgh.harvard.edu/~purcell/plink/; for EIGEN-STRAT, see http://genepath.med.harvard.edu/~reich/Software.htm; for Haploview, see http://www.broadinstitute.org/scientific-community/science/programs/medical-and-population-genetics/haploview/haploview; for WGAViewer, see http://compute1.lsrc.duke.edu/softwares/WGAViewer/; for LocusZoom, see http://csg.sph.umich.edu/locuszoom/; for META, see http://www.stats.ox.ac.uk/~jsliu/meta.html; for SHAPEIT, see http://mathgen.stats.ox.ac.uk/genetics_software/shapeit/shapeit.html; for IMPUTE2, see http://mathgen.stats.ox.ac.uk/impute/impute_v2.html; for the 1000 Genomes Project, see http://www.1000genomes.org/.

## Data availability

The data that support the findings of this study are available from the corresponding authors on reasonable request. The complete GWAS summary statistics are available at http://analysis.bio-x.cn/gwas/ and https://doi.org/10.6084/m9.figshare.7006310.

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

## Acknowledgments

We are deeply grateful to all the study participants, research staff, and students who took part in this work. This work was supported by Chinese National Program on Key Basic Research Project (2017YFA0505900), National Natural Science Foundation of China (Nos. 81101217, 81770006, 81370108, 81330069, 91542111, 31730025, 31325014, 81130022, 81871055, 81701321, 81421061, 21375139, 31571012, and 81501154), Science and Technology Commission of Shanghai Municipality (No. 16ZR1428800), the National Basic Research Program of China (973 Program) (2015CB559100), the National Key R&D Program of China (2016YFC0903402, 2016YFC1306903, 2016YFC0902403, and 2017YFC0908105) and the Program of Shanghai Academic Research Leader (15XD1502200), the National Program for Support of Top-Notch Young Professionals, Shanghai Key Laboratory of Psychotic Disorders (13dz2260500), the "Shu Guang" project supported by the Shanghai Municipal Education Commission and Shanghai Education Development Foundation (12SG17), the Shanghai Jiao Tong Univ Liberal Arts and Sciences Cross-Disciplinary Project (13JCRZ02), Shanghai Hospital Development Center (SHDC12016115), Shanghai Mental Health Center (2016-fx-02).

## Author contributions

B.X.G. and Y.Y.S. conceived and designed the experiments. R.J.Z., Z.Q.L., B.X.G., H.P.L., and S.L.S. reviewed the literature on TB and drafted the manuscript. Y.S., R.J.Z., Z.L., B.X.G., X.F.X., and H.P.L. revised the manuscript. S.L.S. and F.S.H. performed the GWAS SNP chip genotyping. Z.Q.L., J.H.C., J.Z., B.Y.C., Y.H.L., and J.Q.W. performed genotyping of the replication and data analysis. J.Y.C., R.J.Z., Z.Q.L., F.S.H., B.X.G., and Y.Y.S. conducted data analyses. C.H., X.Y.W., H.Y.C., F.L., H.X.W., F.F.X., X.C.C., and X.C.H. recruited samples. J.Y.C., F.L.Z., and S.R.G. generated knockout mice. Y.H.F., L.H.Q., H.Y., Z.H.L., Z.L.C., C.J.Y., J.W., J.H.W., and F.L. performed or contributed to the experiments. All authors critically reviewed and approved the manuscript.

## Additional information

**Competing interests:** The authors declare no competing interests.

