## [Peer Review File · Nature Communications]

Reviewer #1 (Remarks to the Author):

This is an important study that identifies two new TB susceptibility loci through GWA in a Han Chinese population. While relatively small for a GWAS, p-values are genome-wide significant and the additional functional information from PBMCs and mice strengthen confidence. The case-control design appears well done and executed. The genetic association analysis seems well conducted with appropriate correction by PCA with little inflation of lambda. The two genome-wide significant hits are generally convincing (the one in Tgm6 moreso), and the authors are to be commended for including their more exploratory results that did not reach multiple-test corrected significance and comparison of past GWAS hits for TB. I enjoyed reading the study. It is well well-written, succinct, and its findings will be of interest to TB researchers, ID researchers in general, and human geneticists.

Major concerns

- 1) Complete GWAS summary statistics should be deposited in an appropriate database/online location (such as LD Hub) or as a Supplemental Data file to the publication.
- 2) For comparison of mRNAs in Figure 2, it is indicated that the Mann-Whitney U test was used. Isn't this a non-parametric test for 2 samples? Was that the actual test used and some genotypes combined? Or was a different test used? What does simple linear regression with genotypes coded as 0, 1, 2 show? That's more commonly used for testing association with mRNA expression level.
- 3) The section on the Tgm6 KO mouse results could definitely benefit from more detailed description/analysis.
 - a. What are the consequences of the actual CRISPR/Cas9 mutation(s) in the Tgm6 mice? Suppl. Figure 5 does show 110 bp deletion. What are the predicted effects of this? Is there evidence that this is a protein null mutation based on western blot? Where are the probes in the qPCR analysis? Are they prior to exon 4 (indicative of non-sense mediated decay) or do they occur after exon 4, which could indicate there is still normal expression levels of a truncated protein that might have functional consequences. The data are exciting, but the use of a new technology to generate these mice requires careful controls to demonstrate these mice are what the authors were expecting. Furthermore, there should be more experimental detail or at least references to the paper where their method for making KO mice is described? Is this GONAD (Takahashi 2015)?
 - b. What is the phenotype of -/- mice? Is phenotype more severe?
 - c. I'm assuming age/sex matched +/+ littermates are used as controls, but this should be stated explicitly or explained if this is not the case. If het matings are used for propagation, the authors should be able to report the phenotype of all 3 genotypes (unless -/- is lethal).
 - d. In stating the data are "representative" of two experiments in Figure 3, have all the mice been pooled for what is shown or is only the data from one experiment shown, with the implication

that the second experiment showed the same effect? I'd prefer the former, but what has been done should be stated more explicitly.

Minor concerns

None

Reviewer #2 (Remarks to the Author):

Zheng and colleagues present findings from a genome-wide association study of 833 patients with pulmonary tuberculosis and 1220 controls from Shanghai, China. They performed replication of one SNP from each of the top 10 loci (none of which were genome wide significant on their own) amongst two replication cohorts of ~1000 cases and 2000 controls each. They find two variants (rs12437118 and rs6114027) near ESRRB and TGM6 respectively to be associated with TB risk. They go on to show that rs6114027-C is associated with reduced TGM6 expression in healthy controls and TB patients, and that in a mouse model of TB, KO of TGM6 is associated with greater bacterial burden and higher IL-10, TNF- α and CCL3 levels. Understanding of the host genetic correlates of TB is crucial and there have been few prior GWAS studies, and none reported amongst Chinese individuals. Therefore this is a welcome and important contribution to the literature for its studying an important disease in a relatively unique population. Overall the manuscript reads clearly. However with some further additions the manuscript may be more compelling. These findings will be important in the field.

1. With regards the patient cohort, the inclusion criteria stated are either culture confirmed or clinically diagnosed TB cases. Were sputum smear (microscopy) confirmed cases included?
2. With regards genotyping, it has now become common practice to perform genome wide imputation in GWA studies and indeed this has frequently led to novel loci being implicated as well as fine-mapping of associations with greater likelihood of identifying a causal variant. The authors report having done this for the top two loci in the results section. In the methods section it is stated "Un genotyped SNPs within (+/-) 250 kb of the genome-wide significant loci were imputed"; This should be clarified as there were no genome-wide significant loci found it seems. Was there a reason it was not performed genome-wide? Finally, does Figure 1 show the imputed variants as well as genotyped or only genotyped and was the top SNP a genotyped or imputed SNP?
3. It would be helpful if the SNP frequencies were reported both in the text and in Table 1 and Suppl Table 4. One wonders if the allele frequencies may have differed and hence

4. The two top loci may benefit from a bayesian analysis to try to define the credible set of variants; one wonders if the top variant may not actually be the causal variant and hence why the later eQTL analysis and functional assessment is not as revealing as it may be. The use of additional chromatin marks to fine-map is now common-place and could be useful. For example if there are 6 SNPS in the credible set, and only three seem like they are in a genomically active region, I would probably prioritise those. Moreover they may help understand whether ESRRB, whilst being the closest gene may not be the causal gene.
5. It would be helpful if all tables that show OR also show 95% CI (eg Suppl Table 2, Suppl Table 6) for consistency and so that comparisons can be made more easily.
6. The comparison with previous findings is helpful. Was any association seen in the HLA region as was seen in the TB GWAS in Iceland (published in Nature Genetics)? It may be worth imputing the HLA alleles (for example using HIBAG/SNP2HLA/HLA*IMP) and verifying.
7. Analyses of HIV related SNPS seems less valuable than with leprosy related SNPS as it is unclear why HIV should share genetic architecture with TB. The leprosy comparison may be bolstered if coheritability or CPMA type analyses were to be performed.
8. The functional studies are helpful. It may be helpful if the authors reported whether any of the top SNPs are reported in eQTL databases as linked to expression (my search of the blood datasets suggest not). It is therefore a little unclear how the TGM6 eQTL finding be interpreted. Notwithstanding this comment, do the authors have any data that help delineate which cell type within PBMC's may be responsible for the eQTL effect?
9. Further information re the phenotype of the *tgml6* KO mice is important. For example do they show any differences in development, weights at baseline?
10. Were any other cytokines examined in the mouse model prior to selecting IL-10, CCL3 and TNF for presentation?
11. Were there any differences in gross pathology of the mouse lungs in the TGM6 KO vs control mice?
12. I would caution against concluding that *tgml6* is involved in inflammation. It may affect bacterial burden through another means and thereby affect cytokine levels.
13. Can the authors speculate about why these findings were not seen previously? Is it because the allele frequencies in other populations vary? If so these data should be shown.

In summary this is an important and reasonably well conducted study. Like any study it's validity will likely only be assured with further analyses of even larger cohorts but in this area in which we have so few data, this is important and I would support its publication following some modifications.

Reviewer #3 (Remarks to the Author):

The paper by Zheng et al describe a GWAS study in Han Chinese TB patients and describes the identification of two novel risk loci. For one of these locus, TGM6, the authors further show functional insights. The genetic etiology of TB remains to be elucidated, despite evidence that host genetic factors play a role in TB susceptibility. Thus, novel studies adding to this question are important and of interest to the field. This is particularly relevant when large sample sizes are used and functional assays performed to validate associations. The current study fulfils both these important points and is therefore relevant to the field and provides findings from human GWAS to the mouse model of Mycobacterium tuberculosis infection. The study is presented in a clear way, easy to read. In all, the data open new avenues of research to be considered in future TB studies.

There are some points that once addressed would strengthen the work:

- when describing the pathology in *tgm6* WT vs deficient mice, what do the authors mean with «more severe pathological impairments»? This is vague and a proper quantification of the lesions needs to be performed and showed next to the representative histology images.
- what happens at later time points post infection? Does the lack of *tgm6* further exacerbate the infection, or is it compensated in the long term?
- it is not clear the rationale underlying the measurement of IL-10, TNF and CCL3 in the lungs of infected mice. Have the authors tested other cytokines involved in protection in TB, as for example IFN γ ?
- related to the previous point, the authors show that *tgm6*^{+/-} mice show increased bacterial burden and increased expression of a few cytokines. This enhanced expression of TNF, IL-10 and CCL3 could be a result of the increased bacterial burden and not of a direct modulation of inflammatory response by *tgm6*. Thus, it is not clear what hypothesis are the authors discussing.
- considering the observation that *tgm6* expression is lower in the CC genotype and the data from the in vivo infection, it would be interesting to assess if in human cells lower expression of *tgm6* is associated with poorer control of bacteria upon infection and/or with alterations in cytokine expression. Also, is there any clinical evidence pointing to a more severe TB presentation in CC patients (this may be difficult to answer, as it requires access to clinical data, which are not always available; still, if possible it would strengthen the data).
- another functional aspect that is discussed by the authors is the possible participation of *tgm6* in regulating the pattern of cell death induced by M. tuberculosis infection in macrophages. This is an interesting point, not difficult to test, that would significantly increase the functional aspects of the study.

Some points should be addressed in the discussion section, including:

- could differences in ESRRB expression across the different genotypes be restricted to tissue, rather than to PBMCs, considering the role for this molecule in tissue development/regeneration?

- the fact that an ip model of experimental infection was used, when the aerosol route is more physiologic.

Point to point response

We are very grateful for the reviewers' constructive comments. We have revised the manuscript (highlighted in red) to address each of the concerns raised by the reviewers. We hope the revisions will further improve our manuscript and consolidate our findings.

Reviewer #1 (Remarks to the Author):

This is an important study that identifies two new TB susceptibility loci through GWA in a Han Chinese population. While relatively small for a GWAS, p-values are genome-wide significant and the additional functional information from PBMCs and mice strengthen confidence. The case-control design appears well done and executed. The genetic association analysis seems well conducted with appropriate correction by PCA with little inflation of lambda. The two genome-wide significant hits are generally convincing (the one in Tgm6 more so), and the authors are to be commended for including their more exploratory results that did not reach multiple-test corrected significance and comparison of past GWAS hits for TB. I enjoyed reading the study. It is well well-written, succinct, and its findings will be of interest to TB researchers, ID researchers in general, and human geneticists.

We appreciate the reviewer's encouraging and helpful comments.

Major concerns

- 1. Complete GWAS summary statistics should be deposited in an appropriate database/online location (such as LD Hub) or as a Supplemental Data file to the publication.**

The complete GWAS summary statistics will be available at the Bio-X website (<http://analysis.bio-x.cn/gwas/>) after publication of this work, in order to enable researchers to replicate our results and to conduct follow-up researches.

- 2. For comparison of mRNAs in Figure 2, it is indicated that the Mann-Whitney U test was used. Isn't this a non-parametric test for 2 samples? Was that the actual test used and some genotypes combined? Or was a different test used? What does simple linear regression with genotypes coded as 0, 1, 2 show? That's more commonly used for testing association with mRNA expression level.**

We thank the reviewer for the correction. We have reanalyzed the data with linear regression as the reviewer suggested and the statistical data have been included in the revised manuscript.

- 3. The section on the Tgm6 KO mouse results could definitely benefit from more detailed description/analysis. a. What are the consequences of the actual CRISPR/Cas9 mutation(s) in the Tgm6 mice?**

We appreciate the reviewer's advice. More detailed description of the generation of *tgm6* KO mice have been included in the revised manuscript. Briefly, by targeting the exon 4 of *tgm6* gene using the CRISPR/Cas9 gene editing system (1,2), we obtained homozygous *tgm6* knockout mice with a deletion of 110 bp in the genome from 130,136,459 to 130,136,568 on chromosome 2, corresponding to the nucleotides from 299 to 408 in exon 4 of *tgm6* transcript (**Supplementary Figure 6a**).

- 4. Suppl. Figure 5 does show 110 bp deletion. What are the predicted effects of this?**

We thank the reviewer for this question. Neither the mRNA nor the protein of *tgm6* was detected in the lung from *tgm6* deficient mice (**Supplementary Figures 6c-e**), indicating that *tgm6* knockout mice we generated is a protein null mutant.

- 5. Is there evidence that this is a protein null mutation based on western blot?**

We thank the reviewer for the question. By detecting the *tgm6* expression in the lung from wild type and *tgm6* knockout mice by western blot, the abundance of *tgm6* protein was reduced in the heterozygous mice while was undetectable in the homozygous *tgm6* knockout mice (**Supplementary Figure 6e**). Therefore, the *tgm6* knockout mice generated in our study is a protein null mutant.

- 6. Where are the probes in the qPCR analysis? Are they prior to exon 4 (indicative of non-sense mediated decay) or do they occur after exon 4, which could indicate there is still normal expression levels of a truncated protein that might have functional consequences. The data are exciting, but the use of a new technology to generate these mice requires careful controls to demonstrate these mice are what the authors were expecting.**

We appreciate the reviewer's concern. The probes used for the qPCR detection as described in the previous version of the manuscript were designed to target exon 3 of *tgm6*. Moreover, we newly synthesized primers targeting the deleted 110 bp located in the exon 4 of the gene. We observed that the abundance of *tgm6* transcripts as measured by qPCR with these 2 different probes was significantly reduced in the lung from *tgm6* heterozygous mice and was at extremely low level in those from homozygous mice (**Supplementary Figures 6a, c, d**). These data indicating that CRISPR/Cas9 editing of the genome resulted in the non-sense-mediated decay of the mRNA, which is further supported by the fact that the protein of *tgm6* is not detectable in the lung from the homozygous mice (**Supplementary Figure 6e**). Taken together, we generated a *tgm6* protein null mutant mice.

7. Furthermore, there should be more experimental detail or at least references to the paper where their method for making KO mice is described? Is this GONAD (Takahashi 2015)?

We appreciated the reviewer's concern. We generated *tgm6* KO mice by using CRISPR/Cas9 gene editing technology with procedures including embryo isolation, delivery of CRISPR/Cas9 reagents by microinjection and embryo transfer (1,2). The detailed description of the procedures for making the KO mice have been included in the revised manuscript. Genome-editing via Oviductal Nucleic Acids Delivery (GONAD) (3,4) is a novel and simple microinjection-independent genome engineering technique, which can bypass many complex steps in transgenic technology such as isolation of zygotes, microinjection of NAs into them, and their subsequent transfer to pseudo-pregnant animals. However, this technique is not employed in our study.

8. What is the phenotype of -/- mice? Is phenotype more severe?

We appreciate the reviewer's question. We obtained *tgm6* homozygous mice by mating heterozygous mice. The first passages of homozygous mice were used to backcross with wild-type C57/BL6 mice to generate heterozygous offspring and expand the mouse population. We did employ very few homozygous mice (n=2) for the mouse infection experiments before we submitted the manuscript. The results indicated that homozygous (*tgm6*^{-/-}) mice tend to be more susceptible to *Mtb* infection compared with wild type mice manifesting with higher bacterial burdens and exacerbated pathological impairments in the lung (**data not shown**). However, the sample size didn't meet the criteria for statistical analysis and the data is therefore not included in the manuscript. To address the reviewer's question, we have repeated the experiments with wild type, heterozygous and homozygous mice in parallel, the results demonstrated that the tissue damages and bacterial burdens in the lung of *Mtb* infected *tgm6* homozygous mice are severest (**Fig. 3a-d**). These results indicate that the effect of TGM6 on TB pathogenesis is in likely in a dose-dependent manner.

9. I'm assuming age/sex matched +/+ littermates are used as controls, but this should be stated explicitly or explained if this is not the case. If het matings are used for propagation, the authors should be able to report the phenotype of all 3 genotypes (unless -/- is lethal).

We appreciate the reviewer's concern. In all experiments, age/sex matched *tgm6*^{+/+} littermates were used as controls. Heterozygous mating was used for generating homozygous (*tgm6*^{-/-}) mice. The first passages of homozygous mice were then backcrossed with wild-type mice to expand the mouse population. We found that *tgm6* homozygous knockout mice are not lethal (**Supplementary Figure 7a**). There are no apparent differences in the gross phenotype such as body weight among all 3 genotypes (**Supplementary Figures 7a, b**). We have re-performed the animal infection experiments with all 3 genotypes of mice. The

results demonstrated that the pathological impairments and bacterial burden in the lung of *Mtb* infected *tgm6* homozygous mice are severest in comparison with those of wild type and heterozygous mice (**Fig. 3a-d**). These data have been included in the revised manuscript.

- 10. In stating the data are “representative” of two experiments in Figure 3, have all the mice been pooled for what is shown or is only the data from one experiment shown, with the implication that the second experiment showed the same effect? I’d prefer the former, but what has been done should be stated more explicitly.**

We appreciate the reviewer’s concern. The data shown are representative of two independent experiments and the two experiments showed the same effect, but the data were not pooled. We have re-performed the animal experiments with all 3 genotypes, and the pooled data of 4 independent experiments for wild type and heterozygous and of 2 independent experiments for homozygous mice were included in **Fig. 3** of the revised manuscript.

Minor concerns

None

Reviewer #2 (Remarks to the Author):

Zheng and colleagues present findings from a genome-wide association study of 833 patients with pulmonary tuberculosis and 1220 controls from Shanghai, China. They performed replication of one SNP from each of the top 10 loci (none of which were genome wide significant on their own) amongst two replication cohorts of ~1000 cases and 2000 controls each. They find two variants (rs12437118 and rs6114027) near ESRRB and TGM6 respectively to be associated with TB risk. They go on to show that rs6114027-C is associated with reduced TGM6 expression in healthy controls and TB patients, and that in a mouse model of TB, KO of TGM6 is associated with greater bacterial burden and higher IL-10, TNF-a and CCL3 levels. Understanding of the host genetic correlates of TB is crucial and there have been few prior GWAS studies, and none reported amongst Chinese individuals. Therefore this is a welcome and important contribution to the literature for its studying an important disease in a relatively unique population. Overall the manuscript reads clearly. However with some further additions the manuscript may be more compelling. These findings will be important in the field.

We appreciate the reviewer’s encouraging comments and have tried our best to address the reviewer’s concern.

1. **With regards the patient cohort, the inclusion criteria stated are either culture confirmed or clinically diagnosed TB cases. Were sputum smear (microscopy) confirmed cases included?**

We appreciate the reviewer's concern. The inclusion criteria for TB include acid-fast bacilli stain on smear in the first stage of sample collection as mentioned in **Supplementary Figure 8**. The smear and culture-positive cases or smear-negative and culture-positive cases were included, but smear-positive and culture-negative cases were excluded. To avoid confusion, we have revised the description of samples collection in detail in the revised manuscript.

2. **With regards genotyping, it has now become common practice to perform genome wide imputation in GWA studies and indeed this has frequently led to novel loci being implicated as well as fine-mapping of associations with greater likelihood of identifying a causal variant. The authors report having done this for the top two loci in the results section. In the methods section it is stated“ Ungenotyped SNPs within (+/-) 250 kb of the genome-wide significant loci were imputed”; This should be clarified as there were no genome-wide significant loci found it seems. Was there a reason it was not performed genome-wide? Finally, does Figure 1 show the imputed variants as well as genotyped or only genotyped and was the top SNP a genotyped or imputed SNP?**

Thanks so much for pointing these out. We have performed the genome-wide imputation and updated the data accordingly. The **Fig. 1** shows both genotyped and imputed SNPs, and we have indicated the imputed and genotyped SNPs with cross and circle symbols, respectively.

3. **It would be helpful if the SNP frequencies were reported both in the text and in Table 1 and Suppl Table 4. One wonders if the allele frequencies may have differed and hence**

Following the reviewer's suggestion, the SNP frequencies were reported both in the text and in **Table 1** and **Supplementary Table 4**.

4. **The two top loci may benefit from a bayesian analysis to try to define the credible set of variants; one wonders if the top variant may not actually be the causal variant and hence why the later eQTL analysis and functional assessment is not as revealing as it may be. The use of additional chromatin marks to fine-map is now common-place and could be useful. For example if there are 6 SNPS in the credible set, and only three seem like they are in a genomically active region, I would probably prioritise those. Moreover they may help understand whether ESRRB, whilst being the closest gene may not be the causal gene.**

We tried to derive the Bayesian credibility sets (5) for the genome-wide significantly loci (14q24.3 and 20p13) using the results from the discovery stage. However, because sample size of the data set is limited, the 99% credible SNP

sets included over 3000 SNPs and spanned a region of 2 Mb for both of the tested loci. We thus did not show the results in the manuscript. We agree that additional chromatin marks could be useful, and thus annotated the genome-wide significantly SNPs and their linked variants with functional annotations including chromatin state, protein binding and etc. from the Roadmap and ENCODE projects using HaploReg (6). These data were reported both in the text and in **Supplementary Table 7**.

- 5. It would be helpful if all tables that show OR also show 95% CI (eg Suppl Table 2, Suppl Table 6) for consistency and so that comparisons can be made more easily.**

Following the reviewer's suggestion, we have added the 95% CI in all tables that show OR.

- 6. The comparison with previous findings is helpful. Was any association seen in the HLA region as was seen in the TB GWAS in Iceland (published in Nature Genetics)? It may be worth imputing the HLA alleles (for example using HIBAG/SNP2HLA/HLA*IMP) and verifying.**

We thank the reviewer for pointing out this omission. We have investigated the association of the SNPs located in the HLA region (identified in the TB GWAS in Iceland) (7). Two of three SNPs identified in Europeans were replicated in Chinese. Following the reviewer's suggestion, we have imputed the HLA alleles using SNP2HLA, and none genome-wide significant association was observed. The detailed results have been reported both in the text and in **Supplementary Tables 5 and 6**.

- 7. Analyses of HIV related SNPs seems less valuable than with leprosy related SNPs as it is unclear why HIV should share genetic architecture with TB. The leprosy comparison may be bolstered if coheritability or CPMA type analyses were to be performed.**

We agree that it is unclear whether HIV and TB share genetic architecture, and we don't have enough data for the coheritability and CPMA type analyses for leprosy. In addition, our results didn't provide any solid enough evidence in the overlapping analyses, we thus removed these analyses in the revision.

- 8. The functional studies are helpful. It may be helpful if the authors reported whether any of the top SNPs are reported in eQTL databases as linked to expression (my search of the blood datasets suggest not). It is therefore a little unclear how the TGM6 eQTL finding be interpreted. Notwithstanding this comment, do the authors have any data that help delineate which cell type within PBMC's may be responsible for the eQTL effect?**

As the reviewer pointed out, the top SNPs were not found to be linked to expression in the blood datasets. We had also performed the eQTL analysis of the top SNPs in GTEx database and found that *TGM6* gene was not sufficiently

expressed in the whole blood. Real-time qPCR measurement of these genes in PBMCs demonstrated they do express in these cells, though the expression level is very low. Moreover, the SNP rs6114027 were demonstrated to be linked to *TGM6* gene expression in PBMCs (**Fig. 2b**). The promoter reporter assay further showed that such SNP may affect *TGM6* expression by modulating the suppressive effect of the intron on gene transcription (**Supplementary Figure 4c**), which consistently supports the eQTL results. It has been reported that RNA sources from whole blood and PBMCs bears distinct characteristics and most genes were differentially expressed (8), which may be responsible for the discrepancy between the eQTL finding in whole blood and that in PBMCs. Theoretically, the additional cell constituents of whole blood including neutrophils, eosinophils, platelets and reticulocytes (9,10), should not be responsible for the eQTL effect. However, we don't have direct experimental data for delineating which cell type within PBMCs account for the eQTL effect. Below is our eQTL analysis of rs12437118 and rs6114027 in GTEx database.

Table R1. eQTL analysis of rs12437118 and rs6114027 in lung and two other tissues.

Gene Symbol	Gencode Id	SNP	P-Value	Effect Size	T-Statistic	Standard Error	Tissue
ESRRB	ENSG00000119715.10	rs12437118	0.86	-0.022	-0.18	0.120	Cells - EBV-transformed lymphocytes
ESRRB	ENSG00000119715.10	rs12437118	0.77	0.021	0.29	0.074	Lung
ESRRB	ENSG00000119715.10	rs12437118	The gene was not sufficiently expressed in this tissue				Whole Blood
TGM6	ENSG00000166948.5	rs6114027	The gene was not sufficiently expressed in this tissue				Cells - EBV-transformed lymphocytes
TGM6	ENSG00000166948.5	rs6114027	The gene was not sufficiently expressed in this tissue				Lung
TGM6	ENSG00000166948.5	rs6114027	The gene was not sufficiently expressed in this tissue				Whole Blood

The results were extracted from the GTEx project (<https://gtexportal.org/home/>)

9. Further information re the phenotype of the *tgm6* KO mice is important. For example do they show any differences in development, weights at baseline?

We appreciate the reviewer's concern. We monitored the development and body weights of *tgm6* deficient mice. The results of breeding were met with the Mendel's law. Both heterozygous and homozygous mice are viable and show no apparent defects in the body weight and development (**Supplementary Figure 7a**). There is no apparent difference in the gross phenotype among all 3 genotypes at baseline (**Supplementary Figure 7b**).

10. Were any other cytokines examined in the mouse model prior to selecting IL-10, CCL3 and TNF for presentation?

We appreciate the reviewer's question, similar to question #3 raised by reviewer #3. In addition to those altered cytokines and chemokines presented in previous manuscript, we also detected other cytokines including IL-1 β , IL-6, IL-12 and INF- γ , as well as chemokine CCL-2, CXCL-11. However, the expression of these factors was not significantly changed in the lung of both *tgm6* heterozygous and homozygous mice infected with *Mtb* in comparison with those in wild type mice. To better present the data, we have included these results in the revised manuscript (**Fig. 3e**).

11. Were there any differences in gross pathology of the mouse lungs in the TGM6 KO vs control mice?

We thank the reviewer for the question. Histological analysis of the lung by H&E staining showed that *tgm6* deficient mice had more robust infiltration of immune cells and more inflammatory lesions than WT mice at 4 weeks post infection, indicating that *Mtb* infection led to severer tissue damage in the lung of *tgm6* deficient mice. The pathological impairments have been quantified and the data have been included in the revised manuscript (**Fig. 3a, b**).

12. I would caution against concluding that *tgm6* is involved in inflammation. It may affect bacterial burden through another means and thereby affect cytokine levels.

We appreciate the reviewer's concern and agree with the insightful comments similar to the question #4 raised by reviewer #3. As pinpointed by the reviewer, it's highly possible that the enhanced expression of cytokines is secondary to the increased bacterial burden. We didn't have data to support a direct modulatory effect of TGM6 on *Mtb*-induced inflammation. Therefore, following the reviewer's advice, the conclusion have been corrected in the revised manuscript.

13. Can the authors speculate about why these findings were not seen previously? Is it because the allele frequencies in other populations vary? If so these data should be shown.

Following the reviewer's suggestion, we have analyzed the allele frequencies of the newly identified TB associated SNPs (rs12437118 and rs6114027) in other

populations. The results demonstrated that the newly identified SNPs in Chinese also are common in African and European populations (**Supplementary Table 10**). The inconsistency in GWAS may be due to different causal variants or linkage disequilibrium structures in different populations, or phenotypic heterogeneity. Of note, as a complex infectious disease, TB is an outcome of the intricate interaction between genome and environment. The putative environmental influences may result in genome-wide epigenetic modification, which in turn regulates the pathways relevant to TB pathogenesis in cooperation with genetic variation and gene expression. We have commented on this in the discussion part of the revised manuscript.

14. **In summary this is an important and reasonably well conducted study. Like any study it's validity will likely only be assured with further analyses of even larger cohorts but in this area in which we have so few data, this is important and I would support its publication following some modifications.**
We thank the reviewer for the positive comments.

Reviewer #3 (Remarks to the Author):

The paper by Zheng et al describe a GWAS study in Han Chinese TB patients and describes the identification of two novel risk loci. For one of these locus, TGM6, the authors further show functional insights. The genetic etiology of TB remains to be elucidated, despite evidence that host genetic factors play a role in TB susceptibility. Thus, novel studies adding to this question are important and of interest to the field. This is particularly relevant when large sample sizes are used and functional assays performed to validate associations. The current study fulfils both these important points and is therefore relevant to the field and provides findings from human GWAS to the mouse model of Mycobacterium tuberculosis infection. The study is presented in a clear way, easy to read. In all, the data open new avenues of research to be considered in future TB studies.

We thank the reviewer for the encouraging comments.

There are some points that once addressed would strengthen the work:

1. **when describing the pathology in tgm6 WT vs deficient mice, what do the authors mean with «more severe pathological impairments»? This is vague and a proper quantification of the lesions needs to be performed and showed next to the representative histology images.**

We appreciate the reviewer's concern. We determine the pathological impairments in the lung by calculating the region with infiltrated immune cells. As per the reviewer's suggestion, we performed a quantitative analysis of the lesions and calculated the percentage of inflammation area within the whole section of one lung lobe. The updated data have been included in the revised manuscript (**Fig. 3a, b**).

2. What happens at later time points post infection? Does the lack of *tgm6* further exacerbate the infection, or is it compensated in the long term?

We thank the reviewer for this question. The evaluation of longer time points would be helpful to address the reviewer's question. We are sorry that we are not able to provide the data for a time point later than 4 weeks post infection within the limited revision time. However, in our first mice infection experiment, we performed with endpoints at both 2 weeks and 4 weeks post infection. Of note, significant higher bacterial burdens in the lung of *tgm6*^{+/-} mice than those in wild type (*tgm6*^{+/+}) mice infected with *Mtb* for 4 weeks, but not for 2 weeks (**Fig. R1**). These results indicated that the lack of *tgm6* is more prone to further exacerbate the infection, which may not be compensated in the long term of infection.

Fig. R1. The deficiency of *tgm6* increased the bacterial burdens in the lungs of *Mtb*-infected mice. The bacterial burdens in the lungs of wild type (*tgm6*^{+/+}) and *tgm6* deficient (*tgm6*^{+/-}) mice post *Mtb* infection for indicated times were measured by a CFU assay. Data shown are the medians ± interquartile ranges and is from 1 experiment. Each dot represents the data from 1 mice. Mann-Whitney *U* test were used for the statistical analysis. *, *P* < 0.05; ns, not significant.

3. it is not clear the rationale underlying the measurement of IL-10, TNF and CCL3 in the lungs of infected mice. Have the authors tested other cytokines involved in protection in TB, as for example IFN γ ?

We appreciate the reviewer's question, similar to the question #10 raised by reviewer #2. Besides those altered cytokines and chemokines presented in previous manuscript, we also detected other cytokines including IL-1 β , IL-6, IL-12 and INF- γ , as well as chemokine CCL-2, CXCL-11. However, the expression of these factors was not significantly changed in the lung of both *tgm6* heterozygous and homozygous mice infected with *Mtb* in comparison with wild type mice. To better present the data, we have included these results in the revised manuscript (**Fig. 3e**).

- 4. related to the previous point, the authors show that *tgm6*^{+/-} mice show increased bacterial burden and increased expression of a few cytokines. This enhanced expression of TNF, IL-10 and CCL3 could be a result of the increased bacterial burden and not of a direct modulation of inflammatory response by *tgm6*. Thus, it is not clear what hypothesis are the authors discussing.**

We appreciate the reviewer's concern, similar to the question #12 raised by reviewer #2. As pinpointed by the reviewer, it's highly possible that the enhanced expression of cytokines is secondary to the increased bacterial burden. We didn't have data to support a direct modulatory effect of TGM6 on *Mtb*-induced inflammation. Therefore, following the reviewer's advice, we did not conclude the cause and effect relationship of inflammatory response and bacterial burden in the revised manuscript.

- 5. considering the observation that *tgm6* expression is lower in the CC genotype and the data from the in vivo infection, it would be interesting to assess if in human cells lower expression of *tgm6* is associated with poorer control of bacteria upon infection and/or with alterations in cytokine expression. Also, is there any clinical evidence pointing to a more severe TB presentation in CC patients (this may be difficult to answer, as it requires access to clinical data, which are not always available; still, if possible it would strengthen the data).**

According to the reviewer's suggestion, we analyzed the association of *TGM6* SNP rs6114027 with clinical indices including plasma cytokines, erythrocyte sedimentation rate (ESR), the presence of mycobacteria in the sputum smear as well as the occurrence of cavity in the lung of TB patients. Of note, TB patients carrying rs6114027CC genotype showed significantly higher frequencies of mycobacteria in the sputum smear than those with rs6114027TT genotype (**Supplementary Figures 5f**), indicating that the rs6114027C allele is a predisposing factor for severer pulmonary TB disease. We have included the data in the revised manuscript.

- 6. another functional aspect that is discussed by the authors is the possible participation of *tgm6* in regulating the pattern of cell death induced by *M. tuberculosis* infection in macrophages. This is an interesting point, not difficult to test, that would significantly increase the functional aspects of the study.**

As per the reviewer's suggestion, we detected the death of wild type and *tgm6* deficient peritoneal macrophages in response to *Mtb* infection as measured by LDH assay. However, the deficiency of *tgm6* did not significantly affect *Mtb*-induced cell death (**Fig. R2**), indicating the *tgm6* is not involved in the regulation of cell death in macrophages in response to *Mtb* infection. We have changed the claim in the discussion part.

Fig. R2. *Tgm6* didn't significantly affect *Mtb*-induced cell death. LDH assay detection of cell cytotoxicity in wild type (*tgm6*^{+/+}) and *tgm6* deficient (*tgm6*^{-/-}) peritoneal macrophages infected with *Mtb* at a multiplicity of infection (MOI) of 5 for 24 h. Data shown are mean ± SEM and from 3 independent experiments. Statistical differences between groups were analyzed by Student's *t* test. ns, not significant.

Some points should be addressed in the discussion section, including:

- 1. could differences in *ESRRB* expression across the different genotypes be restricted to tissue, rather than to PBMCs, considering the role for this molecule in tissue development/regeneration?**

We appreciate the reviewer's insightful comments. *ESRRB* gene is expressed in all human tissues and play a role in tissue development and regeneration. It's highly likely that the differences in *ESRRB* expression across the different genotypes may be restricted to some other tissue, though no difference was detected between rs12437118 and *ESRRB* mRNA in PBMCs in our study. Following the reviewer's advice, we have commented on this in the revised manuscript.

- 2. the fact that an ip model of experimental infection was used, when the aerosol route is more physiologic.**

We appreciate the reviewer's concern. We fully agree that mouse *Mtb* infection by aerosol route using inhalation exposure system is more physiologic and relevant to natural infection. However, due to the inaccessibility to the aerosol infection model, here we employed an intraperitoneal injection model which has been reported previously (11-13). We have commented on this in the revised manuscript.

References

1. Wang H. *et al.* One-Step Generation of Mice Carrying Mutations in Multiple Genes

- by CRISPR/Cas-Mediated Genome Engineering. *Cell* **153**, 910-918 (2013).
2. Liu W. *et al.* Dosage effects of ZP2 and ZP3 heterozygous mutations cause human infertility. *Hum Genet* **136**, 975-985 (2017).
 3. Takahashi G. *et al.* GONAD: Genome-editing via Oviductal Nucleic Acids Delivery system: a novel microinjection independent genome engineering method in mice. *Sci Rep* **5**, 11406 (2015).
 4. Nagasawa K, Oouchi H, Itoh N, Takahashi KG, Osada M. In Vivo Administration of Scallop GnRH-Like Peptide Influences on Gonad Development in the Yesso Scallop, *Patinopecten yessoensis*. *PLoS One* **10**, e0129571 (2015).
 5. Gaulton KJ. *et al.* Genetic fine mapping and genomic annotation defines causal mechanisms at type 2 diabetes susceptibility loci. *Nat Genet* **47**, 1415-25 (2015).
 6. Ward LD, Kellis M. HaploReg: a resource for exploring chromatin states, conservation, and regulatory motif alterations within sets of genetically linked variants. *Nucleic Acids Res.* **40** (Database issue), D930-4 (2012).
 7. Sveinbjornsson G. *et al.* HLA class II sequence variants influence tuberculosis risk in populations of European ancestry. *Nat Genet.* **48**, 318-22 (2016).
 8. Joehanes R. *et al.* Gene expression analysis of whole blood, peripheral blood mononuclear cells, and lymphoblastoid cell lines from the Framingham Heart Study. *Physiol Genomics* **44**, 59-75 (2012).
 9. Whitney AR. *et al.* Individuality and variation in gene expression patterns in human blood. *Proc Natl Acad Sci U S A.* **100**, 1896-901 (2003).
 10. Min JL. *et al.* Variability of gene expression profiles in human blood and lymphoblastoid cell lines. *BMC Genomics* **11**, 96 (2010).
 11. Sergey Biketov. *et al.* The role of resuscitation promoting factors in pathogenesis and reactivation of Mycobacterium tuberculosis during intra-peritoneal infection in mice. *BMC Infectious Diseases* **7**, 146-146 (2007).
 12. Yang H. *et al.* Lysine acetylation of DosR regulates the hypoxia response of Mycobacterium tuberculosis. *Emerging Microbes & Infections* **7**, 34 (2018).
 13. Wang Y. *et al.* Long noncoding RNA derived from CD244 signaling

epigenetically controls CD8⁺ T-cell immune responses in tuberculosis infection.
Proc Natl Acad Sci U S A. **112**, E3883-92 (2015).

Reviewer #1 (Remarks to the Author):

I appreciate the authors careful and complete responses to the reviewer comments. This work is an important, novel, and convincing contribution to the field, and I have no further criticism to add.

Reviewer #2 (Remarks to the Author):

The authors are to be commended on adequately addressing my comments. Their work is interesting and I hope will advance the field. Whether EDRRB and TGM6 eventually are replicated elsewhere will turn out to be a very interesting thing to watch in the field. I have no further comments.

Reviewer #3 (Remarks to the Author):

The authors have addressed the main issues raised in the first round of revision. They added data, at different levels, and were more cautious about their conclusions, thus making the discussion more balanced. As stated previously, this is a relevant study that combined large sample sizes and functional assays performed to validate associations, including the mouse model of infection. I have no further comments.